# Spatial epidemiology and adaptive targeted sampling to manage the Chagas disease vector *Triatoma dimidiata*

**B. K. M. Case** [1,2] *, **Jean-Gabriel Young** [1,3], **Daniel Penados** [4], **Carlota Monroy** [4], **Laurent Hébert-Dufresne** [1,2], **Lori Stevens** [5]

**1** Vermont Complex Systems Center, University of Vermont, Burlington, Vermont, United States of America, **2** Department of Computer Science, University of Vermont, Burlington, Vermont, United States of America, **3** Department of Mathematics & Statistics, University of Vermont, Burlington, Vermont, United States of America, **4** Laboratorio de Entomología Aplicada y Parasitología, Universidad de San Carlos de Guatemala, Ciudad de Guatemala, Guatemala, **5** Department of Biology, University of Vermont, Burlington, Vermont, United States of America

* bcase@uvm.edu

**Data Availability Statement:** All methods used, as well as results and underlying data shown in our

## Abstract

Widespread application of insecticide remains the primary form of control for Chagas disease in Central America, despite only temporarily reducing domestic levels of the endemic vector *Triatoma dimidiata* and having little long-term impact. Recently, an approach emphasizing community feedback and housing improvements has been shown to yield lasting results. However, the additional resources and personnel required by such an intervention likely hinders its widespread adoption. One solution to this problem would be to target only a subset of houses in a community while still eliminating enough infestations to interrupt disease transfer. Here we develop a sequential sampling framework that adapts to information specific to a community as more houses are visited, thereby allowing us to efficiently find homes with domiciliary vectors while minimizing sampling bias. The method fits Bayesian geostatistical models to make spatially informed predictions, while gradually transitioning from prioritizing houses based on prediction uncertainty to targeting houses with a high risk of infestation. A key feature of the method is the use of a single exploration parameter, $\alpha$, to control the rate of transition between these two design targets. In a simulation study using empirical data from five villages in southeastern Guatemala, we test our method using a range of values for $\alpha$, and find it can consistently select fewer homes than random sampling, while still bringing the village infestation rate below a given threshold. We further find that when additional socioeconomic information is available, much larger savings are possible, but that meeting the target infestation rate is less consistent, particularly among the less exploratory strategies. Our results suggest new options for implementing long-term *T. dimidiata* control.

figures, are available at https://doi.org/10.5281/zenodo.6462160.

**Funding:** BC, LHD and JGY acknowledge support from the National Institutes of Health 1P20 GM125498-01 Centers of Biomedical Research Excellence Award. BC is also supported as a Fellow of the National Science Foundation under NRT award DGE-1735316, and LHD by the National Science Foundation award EPS-2019470. The field data used in this work was supported by the International Development Research center of Canada (IDRC subsidy no. 106531) awarded to CM. The funders had no role in study design, data analysis, decision to publish, or preparation of the manuscript.

**Competing interests:** The authors have declared that no competing interests exist.

## Author summary

Effective public health interventions for the control and elimination of neglected tropical diseases require an efficient use of resources while still causing long-term disease reduction at the community level. To use resources to best effect, areas most in need of control efforts must be identified. However, strategies for correctly identifying these areas are rarely known due to the complex environmental, biological, and cultural factors shaping disease spread. In turn, incorrect prioritization of control targets can cause the intervention to have no lasting effect. We address this tradeoff between efficiency and efficacy by adapting control priorities throughout an intervention, targeting areas of high uncertainty during the initial stages while shifting to areas of greatest risk at later stages. In the context of controlling *Triatoma dimidiata*, the primary vector of Chagas disease in several countries in Latin America, our methods provide a means of targeting only a subset of homes for insecticide and housing improvements, while still reducing a village's overall infestation rate below the critical threshold.

## 1 Introduction

Chagas disease is a vector-borne neglected tropical disease (NTD) endemic to all countries in Latin America [1]. It is the most serious parasitic disease in the region, with a 2005 estimate of disease burden 5 to 10 times greater than malaria [2], and is mainly a threat to people living in poverty [3, 4]. The disease, which can lead to potentially fatal cardiovascular or gastrointestinal issues, is caused by the parasite *Trypanosoma cruzi* and transmitted by insects in the Triatominae subfamily [5]. Control initiatives for Chagas primarily focus on disrupting the transmission pathway to humans by reducing domestic Triatomine infestation levels, which is the primary mode of infection [6, 7]. A common control target is to reduce the proportion of infested households in a community to below 5% [8, 9], and it has been shown that reduction past 8% is sufficient to eliminate *T. cruzi* seroprevalence in children aged 6 months to 15 years [10].

In Central America, the prevalence of Chagas disease has declined significantly since the 1980s, thanks in part to the near-elimination of the invasive vector *Rhodnius prolixus*. However, the species-complex *Triatoma dimidiata* still poses a significant health risk to millions of people in many areas [11]. Unlike the invasive *R. prolixus*, *T. dimidiata* is endemic to Central America, living in peridomestic and sylvatic as well as domestic environments [12]. Unfortunately, efforts to control domestic *T. dimidiata* populations are complicated by the resilience of *T. dimidiata* to traditional methods of vector control. Domestic populations of the insect can rebound within several months of insecticide spraying [13–15], and continuing to spray houses which report reinfestation appears to have little long-term effect [16]. Control measures for *T. dimidiata* are further complicated due to its significant variation in habitat, morphology, feeding patterns, and genetics [17–19], all of which interact to cause variation in its vulnerability to insecticides [20] and the domiciliary risk factors associated with its presence [9, 21]. Further, the sustainability of a given control strategy depends critically on cultural practices in the area [22–24]. Thus, meeting the goal of long-term *T. dimidiata* reduction requires adaptive, locale-specific strategies for surveillance and control [25, 26].

The limitations of insecticide for control of *T. dimidiata* in Guatemala have led to the gradual adoption of additional measures, mostly in the departments of Jutiapa and Chiquimula in southeastern Guatemala, which represent the majority of the country's cases reported to the Ministry of Health [11, 27]. A promising multidisciplinary approach, often referred to as the

EcoHealth approach, applies cost-effective, locally-tailored house and peridomestic improvements by collaborating with villagers and health personnel, in conjunction with initial insecticide application [22]. A pilot study of two villages found the method led to very low ($<$ 5%) infestation rates 5 years after housing improvements [21, 28], and an expansion of the project to five villages in Chiquimula led to a sustained four-fold reduction in infestation [25]. This suggests house improvements following the EcoHealth approach can effectively prevent reinfestation in the long term.

Barriers to the widespread adoption of community engagement-based interventions include frequent shortages in insecticides [25] and the need for research experts and trained personnel to work with residents and identify infested houses. One possible solution to these issues is to more efficiently select homes in need of insecticides and improvements while still meeting necessary control targets. For example, by treating only a sample of homes such that overall the *T. dimidiata* infestation rate in the village goes below 5%, residual dispersal and non-domestic migration will be limited to homes that were recently improved, or were already unlikely to be suitable for infestation. Further, the EcoHealth approach's emphasis on practical improvements and community participation could help ensure that the risk factors identified from this sample continue to be addressed throughout the entire village. This control strategy would in turn free up resources to be applied in other communities.

To be successful, such a strategy must balance the incentive to target houses that are believed to be infested, with the need to correctly identify the infestation status of unvisited homes. Selecting houses to treat based on perceived infestation risk will quickly bring the village's infestation rate closer to the 5% goal, at least in the short term. However, a diverse range of samples throughout the area and across combinations of possible risk factors is required to reliably find the remaining infested houses, and to correctly predict whether the remaining number of infested houses is below that required to meet the 5% threshold [9, 29]. This was the conclusion in King et al. (2011), where a subset of households was inspected for Triatomines in villages across Guatemala using either random sampling or sampling based on pre-defined risk factors, and was used to predict whether the village infestation rate was below or above the 5% threshold. The authors found random sampling to consistently have higher prediction accuracy, noting that sampling based on a fixed set of factors failed to explore the ways factors associated with infestation vary among different villages and regions [9]. In short, the dual objectives of prediction and targeted sampling lead to a exploration vs. exploitation trade-off, where the space of houses to target must be searched to find configurations that are both of minimal sample size, and which contain enough information to correctly predict that the 5% target has been met.

The quantity and quality of covariate information available for exploration is a second key factor in the success of an intervention strategy. A number of studies have identified various socioeconomic factors associated with infested houses, such as the material and condition of house walls [4, 21, 30]. Therefore, if additional dependent variables are available prior to selecting houses for treatment, fewer observations may be needed to make accurate estimates. Another option is to rely only on variables available remotely, such as elevation. While this sacrifices potentially useful information, it may ultimately be more cost effective, since collecting socioeconomic risk factors for inference and prediction requires additional labor and logistical planning [31].

Here we aim to address the problem of treating a subset of houses to reduce *T. dimidiata* infestation to a target threshold while minimizing the necessary resources. Using 5 villages of varying size and baseline infestation rates in Chiquimula, Guatemala as a case study, we employ *adaptive geostatistical design* strategies which sequentially select houses based on

observations from previous iterations, and use inherent spatial autocorrelation in the observed data to improve prediction and inference.

While historically quite theoretically driven, the principles of geostatistical design have recently been applied to other problems of survey design and analysis in spatial epidemiology [32]. Chipeta et al. (2016) developed an adaptive sampling method which targets locations with high spatial uncertainty, and applied the approach to a cross-sectional malaria survey [33, 34]. Adaptive sampling using a similar strategy based on prediction entropy and spatial exploration was also shown to be effective for identifying hotspots of lymphatic filariasis [35]. Fronterre et al. (2020) used a non-adaptive, lattice-like sampling design combined with close pairs of points (proposed first in [36]) to predict whether an area's disease prevalence exceeds a certain threshold, and found the method outperformed a current WHO assessment protocol on a simulated dataset [37].

In this work, we develop a class of adaptive strategies which transition from prioritizing houses based on prediction uncertainty to houses based on percieved risk of infestation. We compare these strategies to random sampling with empirical data, and assess their ability to efficiently locate infested houses while correctly predicting whether the current selection meets the reduction target. Additionally, we examine the effect of including socioeconomic covariates on the performance of each strategy. In the context of Chagas vector control with the EcoHealth approach, our methods address two key questions: 1) how can houses be more efficiently targeted for treatment to sufficiently reduce village-wide vector incidence? and 2) can further efficiency gains be made by collecting additional socioeconomic information? More generally, our methods provide a formal, statistical framework for targeted control strategies in a resource-limited setting, and hence are particularly relevant to the control of NTDs.

## 2 Methods

### 2.1 Data preparation

Our data come from the follow-up EcoHealth project discussed in the introduction, which was conducted in the countries Honduras, El Salvador, and Guatemala, between August and December 2011, and is described in detail in Bustamante et al. (2015) and Lima-Cordón et al. (2018) [18, 21]. We focus only on the five villages in Chiquimula, Guatemala, since infestation rates were low in the other countries. These villages lie along an altitudinal gradient, with a climate ranging from hot and humid to cooler cloud forest. Villages are surrounded by a mix of banana plantations, shade grown coffee, and patches of the original forest [18].

All houses with missing factors necessary for our analyses were removed, leaving between 72% and 83% of the total number of houses recorded in each village (S1 Table). After processing, there were 172 housing structures in El Amatillo, 147 in El Cerrón, 251 in El Guayabo, 108 in El Paternito, and 207 in La Prensa, for a total of 885 observations. The village-wide infestation rate was between 15% and 39%.

Each data entry was obtained by two trained personnel using the following protocol. After the informed consent of the residents, houses were searched for 35–45 minutes by one team member with a flashlight and forceps, searching walls, behind furniture, and other suitable environments for Triatomine shelter, while another performed interviews and assessed aspects of tidiness in the home [21, 28]. The home's geocoordinates were also recorded. These surveys produce a binary response indicating Triatomine presence in the home, and 26 covariates. A positive response indicates that adult or juvenile insects, dead insects, or eggs were found. The covariates, listed in S2 Table, include items related to socioeconomic factors, the number and type of domestic animals, and the house's structure and cleanliness. We added two more covariates based on the house's geocoordinates. The *distance to perimeter* is the shortest

distance of the home to the village's convex hull, plus 50m, while the *density* is the number of other houses within 100m. Covariates were checked for multicollinearity, and all continuous variables were centered and scaled. Additionally, for convenience in setting priors, the coordinates of the houses were scaled such that the diameter of the village (maximum distance between any two points) was one.

## 2.2 Hierarchical modeling for geostatistics

*Geostatistics* is a field which studies spatial autocorrelation in point-referenced data, and leverages this information for inference and prediction [38]. Geostatistical models incorporate a spatial phenomena $Z = \{z(\boldsymbol{s}) \in \mathbb{R} \mid \boldsymbol{s} \in \mathscr{D}\}$ over a domain of possible locations $\mathscr{D}$, where $n = |\mathscr{D}|$ when $\mathscr{D}$ is discrete. The closer two points $\boldsymbol{s}_i$ and $\boldsymbol{s}_j$ are to each other, the more similar the values $z(\boldsymbol{s}_i)$ and $z(\boldsymbol{s}_j)$ will tend to be. This spatial surface $Z$ is itself a function of possibly unknown *spatial parameters*, which control how the covariance between points behaves. Rather than observing $Z$ directly, for each location $\boldsymbol{s}$ there is typically a measurable response $y(\boldsymbol{s})$, which is assumed to be a function of $Z$ and some covariates $\boldsymbol{x}(\boldsymbol{s}) = x_1(\boldsymbol{s}), \ldots, x_p(\boldsymbol{s})$.

Following the general hierarchical framework first outlined in [39], we assume the response at each $\boldsymbol{s} \in \mathscr{D}$ follows a generalized linear model with spatially correlated random effects. In our setting, this amounts to $y(\boldsymbol{s})$ being a binary variable indicating the infestation status of a home at position $\boldsymbol{s}$, which has a probability $r(\boldsymbol{s})$ of being infested. The probability $\sigma_s$, or the *risk* of having an infestation at location $\boldsymbol{s}$, will then depend on the home's covariates and the risks of other homes in close proximity. More formally, the likelihood of $y(\boldsymbol{s})$ is as follows:

$$y(\boldsymbol{s}) \mid r(\boldsymbol{s}) \sim \text{Bernoulli}(r(\boldsymbol{s})) \tag{1}$$

$$\text{logit}(r(\boldsymbol{s})) = \eta(\boldsymbol{s}) = \boldsymbol{x}(\boldsymbol{s})\boldsymbol{\beta} + z(\boldsymbol{s}) + \boldsymbol{\varepsilon}(\boldsymbol{s}), \tag{2}$$

where $\boldsymbol{\beta} = (\beta_1, \ldots, \beta_p)^\top$ is a vector of fixed effect coefficients. The spatial surface $Z$ follows a zero-centered Gaussian distribution with Matérn covariance function (defined in S1 Appendix) with smoothness parameter $\nu = 1$ [40]. This spatial process has two parameters $\sigma_s$ and $\rho$, which respectively control the variance and *effective range*, here defined as the distance at which the correlation between two points reaches 0.1. Finally, $\varepsilon$ is an independent random effect representing non-spatial latent variability at each location, which helps avoid finding spurious spatial correlation [37]. We assume $\varepsilon \sim N(0, \sigma_e^2)$.

The equations above describe the probability a house is infested, given some parameters and covariate information. In other words, they specify an assumption about how our response is generated as a function of these parameters. However, we are interested in reversing this process: given some finite set $\mathscr{S} \subset \mathscr{D}$ of locations, we will use the household data from these locations to make inferences about possible values of the parameters. Following convention, we write $\boldsymbol{y} = \{y(\boldsymbol{s})|\boldsymbol{s} \in \mathscr{S}\}$ as the observed response, and $\boldsymbol{\theta} = (\boldsymbol{\beta}, \rho, \sigma_s, \sigma_e)$ as the parameters to be estimated. Under the Bayesian paradigm, one treats $\boldsymbol{\theta}$ as a random variable and assigns priors based on domain knowledge or on hypotheses formed prior to data collection. Bayes' theorem then gives the posterior distribution $p(\boldsymbol{\theta}|\boldsymbol{y})$, which represents our updated beliefs about $\boldsymbol{\theta}$ given the data we have observed at $\mathscr{S}$. The posterior distribution can further be used to make predictions at unmeasured locations.

For our analysis, we use weakly-informative $N(0, 3.3)$ priors for $\boldsymbol{\beta}$, while for the spatial range and standard deviation we use the penalized-complexity prior of [41], set to induce tail probabilities of $\Pr(\rho < 0.1) = 0.05$ and $\Pr(\sigma_s > 3) = 0.1$. The variance for the non-spatial random effects $\sigma_e^2$ follows an inverse-gamma distribution with location 1 and scale 0.01.

Data preparation and analysis was performed in R version 4.1.0 [42]. For computational speed and convenience, all statistical models were fit using Integrated Nested Laplace Approximation (INLA) with the Stochastic Partial Differential Equation (SPDE) representation for the spatial effects, available from the R-INLA package [43, 44]. All materials necessary for the analysis are publicly available online [45], including a brief tutorial on spatial modeling with INLA.

## 2.3 Model comparison and full-village analysis

To verify the suitability of the model outlined above, we compare its performance to two simpler alternatives. The first removes the correlated spatial random effects $Z(\boldsymbol{s})$ while the independent $\varepsilon(\boldsymbol{s})$ effects are removed from the other, but otherwise each model is the same. Models are evaluated based their *deviance information criterion* (DIC) and *marginal likelihood* (ML), formally defined in S1 Appendix. Both measure a model's goodness-of-fit to the data while penalizing model complexity.

Each village is analyzed separately, using all available data within the village. Additionally, we consider two sets of covariates to be available. The *global* covariate set contains only variables obtainable from the geocoordinates, namely, the location's density and distance to the perimeter, which represents a small amount covariate information that is convenient to collect. The other set contains these variables along with the socio-economic covariates listed in S2 Table.

## 2.4 Predicting out-of-sample infestation status

Let $\boldsymbol{y}_0 = \{y(\boldsymbol{s}_0)|\boldsymbol{s}_0 \in \mathscr{D}\backslash\mathscr{S}\}$ indicate the unknown infestation status at unvisited locations. Given a response $\boldsymbol{y}$ observed at $\mathscr{S}$, the joint posterior predictive distribution for $\boldsymbol{y}_0$ is then

$$\Pr\left(\hat{\boldsymbol{y}}_0 \mid \boldsymbol{y}\right) = \int \Pr\left(\hat{\boldsymbol{y}}_0 \mid \boldsymbol{y}, \boldsymbol{\theta}\right) p\left(\boldsymbol{\theta}|\boldsymbol{y}\right) d\boldsymbol{\theta}. \tag{3}$$

We are interested in the total count of unvisited locations which are infested, defined as $I_0 = \mathbf{1} \cdot \boldsymbol{y}_0^\top$. To estimate the distribution $\Pr\left(\hat{\boldsymbol{y}}_0 \mid \boldsymbol{y}\right)$, and hence $\Pr\left(\hat{I}_0 \mid \boldsymbol{y}\right)$, from the posterior, we generate 5,000 Monte Carlo samples from $p\left(\boldsymbol{\theta}|\boldsymbol{y}\right)$, then for each of these samples $\boldsymbol{\theta}^{(i)}$, we draw a sample from $\Pr\left(\hat{\boldsymbol{y}}_0 \mid \boldsymbol{y}, \boldsymbol{\theta}^{(i)}\right)$, resulting in 5,000 samples from (3).

## 2.5 An adaptive sampling strategy for infestation reduction

In the present study, our objective is not only to make predictions given data observed at $\mathscr{S} \subset \mathscr{D}$, but to choose the set $\mathscr{S}$ itself to best control infestation. In this context, $\mathscr{S}$ is referred to as the *sampling design*. To evaluate the quality of a sampling design, we specify a *design target*. In our context, this target is the number of houses selected for treatment (i.e. the size of $\mathscr{S}$), subject to the constraint that the true infestation rate among unvisited houses is below 5% (i.e. $I_0/n < 0.05$).

One possible strategy for choosing effective sampling designs is adaptive (or sequential) sampling. Rather than specifying our set of houses to sample completely before collecting data at these houses, we sample houses in batches: a collection of houses is selected, the data is gathered at these houses, a model is fit to the data so far, and a new batch of houses is chosen based on the model's predictions. At the end of this process, the set of houses we have visited becomes our final sampling design.

Implementing our adaptive strategy requires the following [33]: 1) an initial design $\mathscr{S}_1$ from which to fit the first model, 2) a *batch size b* as the number of new houses to add to the existing observations each iteration, 3) a *utility function U* to rank unobserved houses to target.

For our application, we additionally require 4) a *termination condition* to predict whether the current design meets the infestation target.

For the utility function, a natural choice might be to rank a location $s_0 \in \mathcal{D} \backslash \mathcal{S}$ according to its posterior predicted risk $\Pr(\hat{y}(s_0) = 1 \mid y) = E[r(s_0) \mid y]$. However, this strategy has high sampling bias and fails to prioritize locations with potentially new information, hence ignoring possibly large amounts of the geospatial or covariate search space.

To better explore this space, we therefore balance this strategy with prioritizing locations of high uncertainty. Since predictions are generally less reliable with smaller sample sizes, the proposed utility function transitions smoothly from prioritizing a location's variance $\tilde{v}(s_0)$, to prioritizing its expected risk $\tilde{r}(s_0)$, where $\tilde{v}(s_0)$ and $\tilde{r}(s_0)$ have each been centered and scaled over the unvisited locations. If $m_i$ is the current number of locations observed at iteration $i$ of sampling, we rank locations according to the utility function

$$U(s_0) = t(i; \alpha)\tilde{r}(s_0) + (1 - t(i; \alpha))\tilde{v}(s_0) \tag{4}$$

where $t(i; \alpha) = ((m_i - m_1)/(n - m_1))^\alpha$ is a weighting function interpolating between 0 and 1. The *exploration parameter* $\alpha > 0$ controls the speed at which the predicted risk is prioritized, with $\alpha < 1$ transitioning to risk-based targeting more quickly and $\alpha > 1$ favoring exploration for longer.

We propose a termination condition based on our confidence that the current design satisfies the reduction target:

$$\Pr(\hat{I}_0 < \kappa n) \geq \gamma, \tag{5}$$

where $\kappa$ is the desired infestation rate, and $\gamma$ the desired confidence level. If $\mathcal{S}_i$ is the current design, we can compute this probability using draws from $\Pr(\hat{I}_0 \mid y_i)$ as described above. We fix $\kappa = 0.05$ and $\gamma = 0.95$ throughout the manuscript.

Finally, we set the batch size $b = 3$, and sample the initial design $\mathcal{S}_1$ uniformly at random. In summary, the adaptive sampling algorithm is as follows.

1. Sample initial design $\mathcal{S}_1$ uniformly and set $i = 1$

2. Fit the posterior distribution $p(\theta \mid y_i)$ as described in Section 2.2

3. If $\mathcal{D} \backslash \mathcal{S}_i$ satisfies (5), return $\mathcal{S}_i$. Otherwise, go to step 4

4. Assign each location $s_0 \in \mathcal{D} \backslash \mathcal{S}_i$ a utility $U(s_0)$ using (4)

5. Choose the $b$ locations with highest utility and add them to $\mathcal{S}_i$

6. Set $i = i + 1$ and repeat steps 2–6.

## 2.6 Simulation study comparing adaptive and random sampling

We compare the performance of the adaptive sampling procedure from above using several values of the exploration parameter $\alpha$, along with a random sampling procedure, on each of the five Guatemalan villages. To evaluate each resulting design, we record the size $m$ of the final design, and the true infestation rate, defined as the number of truly infested houses not in the design divided by the total number of houses in the village. For all experiments, we used a batch size $b = 3$.

For each village, the following experiment is repeated 50 times. An initial set of 10 locations is chosen at random. Then, adaptive sampling using each of $\alpha = 0, 0.15, 0.3, 0.7, 1, 2$ is performed, as well as a random procedure which uniformly samples $b$ new houses each iteration,

resulting in 7 final designs to be evaluated. In this way, we apply each of the different proce-
dures to the same set of randomized initial samples.

## 3 Results

### 3.1 Full-village analysis

A comparison of the model given in Section 2.2 to the two simpler alternatives is shown in
Fig 1 for each village. The proposed model had a lower (better) DIC than the alternatives in all
five villages, except for El Cerrón with the full covariate set, where the model with the spatial
effects removed did slightly better. The spatial-only model ($\varepsilon(s)$ removed) had higher ML in all
but 3 cases. However, the two models including a spatial effect had similar performance
according to both measures. In contrast, in El Guayabo, El Paternito, and La Prensa, these two
models had substantially better performance than the model with spatial effects removed. Fig 1
also shows the computation time of the proposed model was considerably faster than the spa-
tial-only model, while the model with no spatial effects was fastest.

The proposed model, fit to all data in the village, also allowed us to draw inferences about
the nature of *T. dimidiata* infestation at a local level. Table 1 summarizes the effective range
after rescaling back to meters, while S1 Fig shows the full posterior distribution. The posterior
mode was between 150m and 874m, or between 10% and 40% of the village diameter.
Although there was notable variation in the effective range between the villages, it appears
fairly similar between the two covariate sets for a given village, with a difference in mode
between 6m and 223m. In the villages El Cerrón and El Paternito, the 95% highest posterior

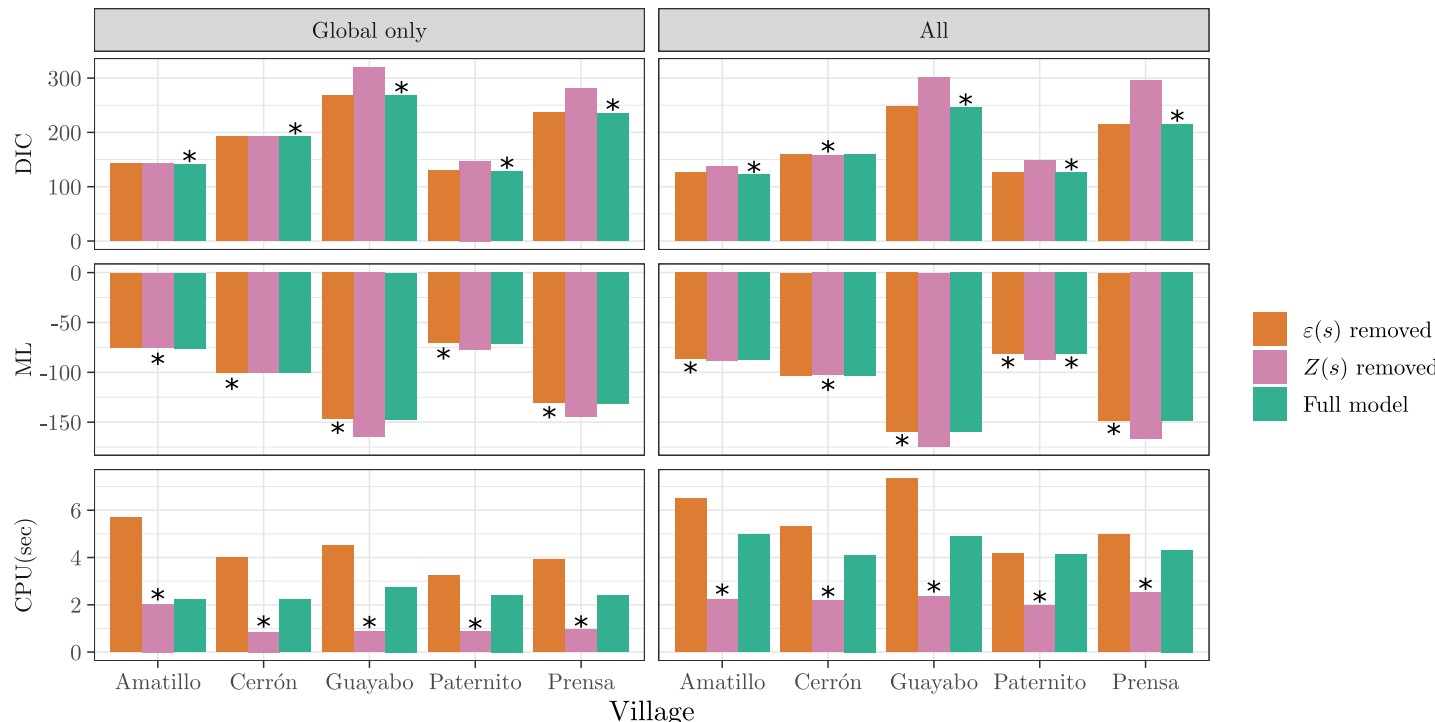

**Fig 1. Model comparison based on two goodness-of-fit measures and computation time, by village and level of covariate information.** The best-performing model
is indicated with a star in each case. "Global only" means only covariates available from the coordinates are available, and "All" indicates all 28 covariates were used. "$Z(s)$
removed" refers to the model outlined in Sec. 2.2 but with no spatial effects, "$\varepsilon(s)$ removed" refers to the independent effects removed, and "Full model" refers to the full
model. DIC and ML are defined in S1 Appendix.

**Table 1. Posterior of the effective range from the full-village analysis.** Each posterior is fit according to the model of Section 2.2, using two different sets of covariate information. The 95% highest posterior density interval (HPDI) is the smallest interval such that 95% of the posterior mass is contained within it.

| Village | Covariate set | Mode & 95% HPDI (meters) |
|---------|---------------|--------------------------|
| El Amatillo | Global only | 156 (37–931) |
|  | All | 150 (62–420) |
| El Cerrón | Global only | 728 (4–8201) |
|  | All | 553 (6–7370) |
| El Guayabo | Global only | 774 (334–1910) |
|  | All | 823 (358–2062) |
| El Paternito | Global only | 651 (161–2860) |
|  | All | 874 (171–3751) |
| La Prensa | Global only | 480 (212–1138) |
|  | All | 508 (205–1281) |

density interval (HPDI) for the effective range was quite large (spanning several kilometers), which suggests the spatial signal was weakest in these villages.

Examining the coefficients for the fixed effects, we found variation between the villages in the majority of the covariates in the full covariate set (Fig 2). The factors which consistently had a negative association with infestation, defined here as having a 50% HPDI fully below 0, were not having rats in the home (5 villages), having bedroom walls in good condition (4 villages), and not keeping construction materials around the home (4 villages). The factors with a consistent positive association were having a kitchen outside the home (4 villages), evidence of bird nests inside (3 villages), and keeping chickens outside the home (3 villages).

## 3.2 Simulation study results

The results of the simulation experiment comparing adaptive and random sampling strategies are shown in Fig 3. From the 50 final designs obtained from each group of strategy × village × covariate set, we calculated the mean and 90% confidence interval (CI) for the percentage of houses in the final design, and for the true infestation rate remaining in the village. In all villages and both covariate sets, all adaptive strategies had a smaller mean design size (final number of sampled houses) than random. Moreover, when the exploration parameter was less than $\alpha = 2$, the 90% CI for the design size was entirely below (i.e. not overlapping) the CI for random sampling. Comparing the top and bottom rows of Fig 3, we further find that, except for when $\alpha = 2$ and random sampling was used, having the full covariate set tended to lead to a smaller design size compared to having the global covariate set.

Although the adaptive strategies tended to produce smaller designs, they were also more likely to produce designs which failed to satisfy the design target of reducing the true infestation rate below 5%, especially when the exploration parameter $\alpha$ was too low. This highlights the importance of exploring the design space during early stages of sampling, in order to mitigate prediction bias. With the global covariate set, in four villages all adaptive strategies had a 90% confidence interval below the target threshold, i.e. at least 95% of the designs had a true infestation rate below 5% in the final sample, while in La Prensa only $\alpha \geq 0.7$ gave a CI below the target. With the full covariate set, the strategies with lower $\alpha$ missed the target threshold more frequently. The mean infestation rate was above 5% for $\alpha \leq 0.3$ in two villages, and in one village for $\alpha = 0.7$. Moreover, the CI contained 5% in four villages for $\alpha \leq 1$, and one village for $\alpha = 2$. The CI was always below 5% with random sampling.

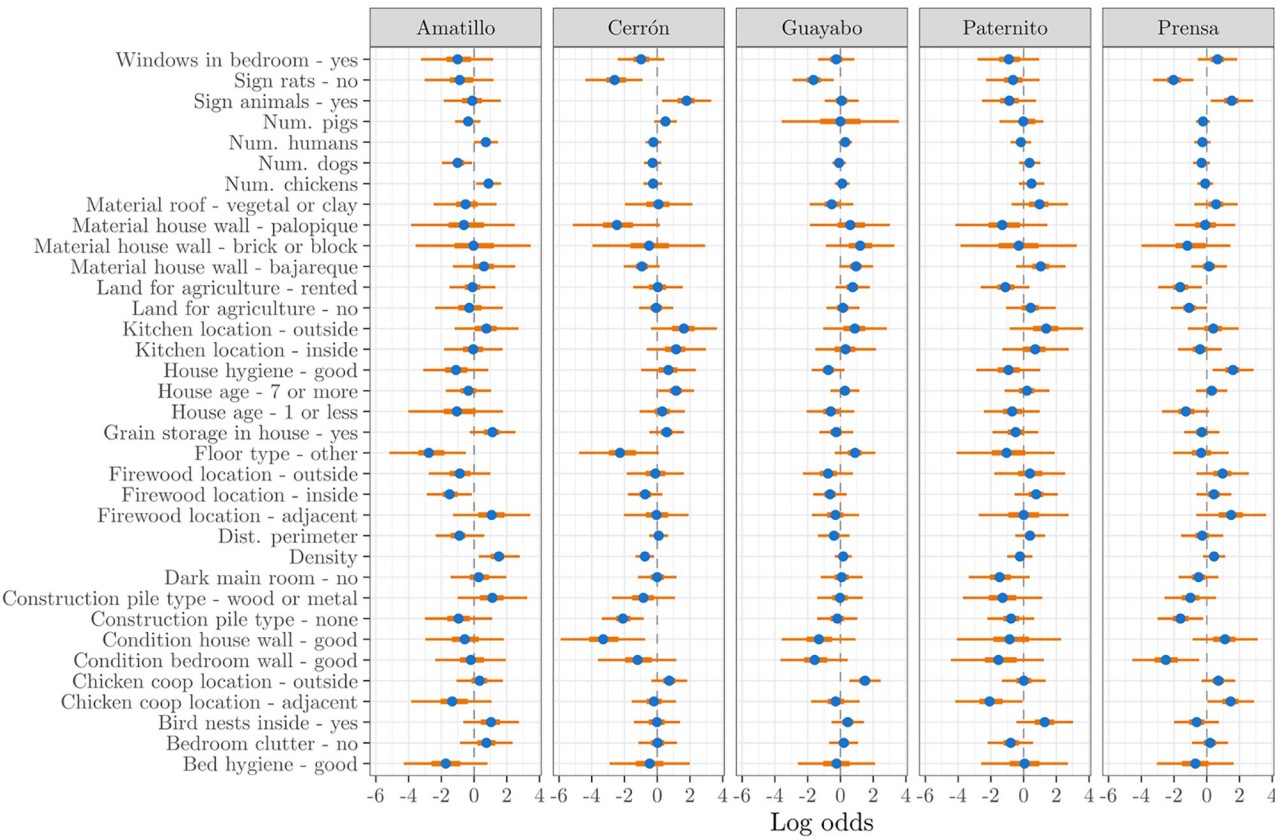

**Fig 2. Fixed effect coefficients from fitting the model of Section 2.2 to all available data in each village.** Coefficients are on log odds scale. The blue point is the posterior mean, the bold orange line is the 50% HPDI, and the thin orange line is the 95% HPDI.

Table 2 provides a higher-level perspective of the relative performance of the adaptive strategies, with results from all villages pooled together. Again, we find that the accuracy (percentage of designs which met the 5% infestation target) was lowest for lower values of $\alpha$, especially with the full covariate effect, and that accuracy was inversely proportional to the difference in design size compared to random.

## 4 Discussion

The resilience of NTDs, combined with the resource-limited context in which they inherently reside, presents unique challenges for their control and elimination. Interventions must not only provide effective and long-term disruption of the disease, but also be logistically feasible to allow widespread application [46]. This leads to a difficult tradeoff between preserving resources while actually making a difference at the community level. Because of the complex environmental, ecological, and cultural aspects of NTDs, a further challenge is the need for spatially and temporally adaptive solutions for efficient control [26, 47].

We have proposed an algorithm which balances this tradeoff between efficiency and efficacy in the context of Chagas disease vector control in southeastern Guatemala. With the goal of reducing a village's domiciliary Triatomine infestation rate below the government target of

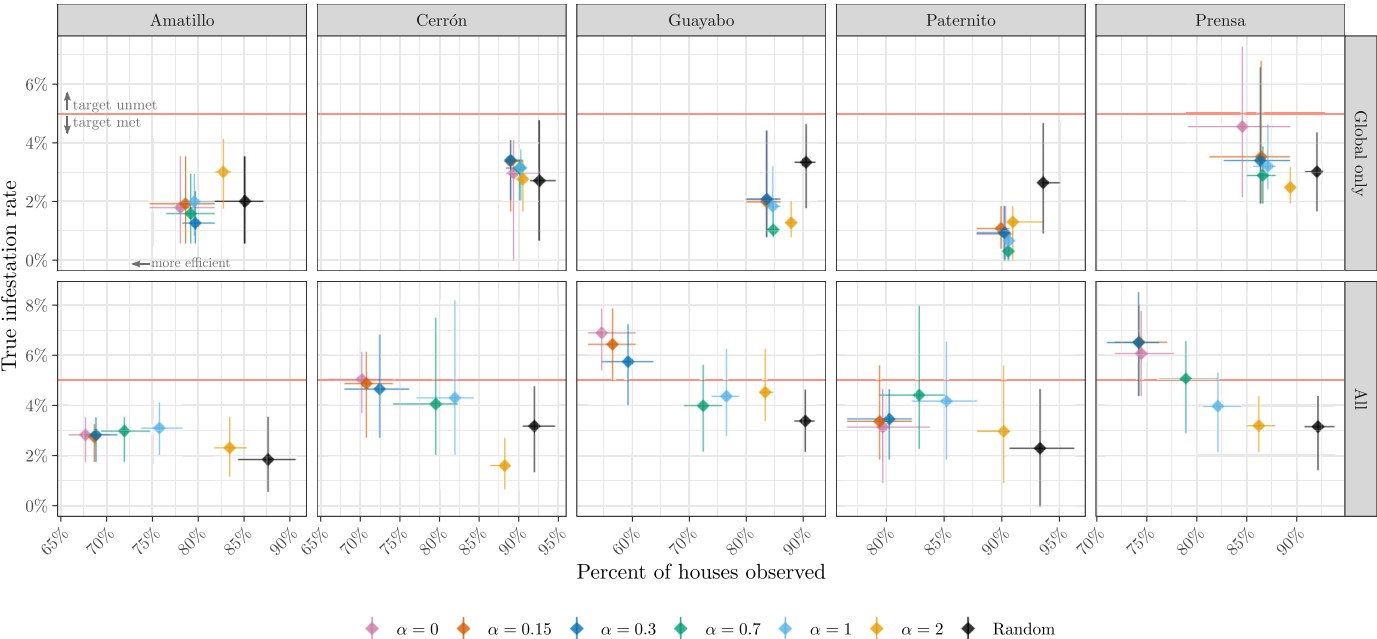

**Fig 3. Results from the simulation experiment for each village and predictor set.** The adaptive sampling procedure for varying $\alpha$, along with random sampling, is performed for 50 initial sets of 10 houses. The x-axis is the percentage of houses in the final sampling design, and the y-axis is the true infestation rate remaining in the village. Diamonds show means, while cross-hairs show the 90% confidence intervals of the data corresponding to each axis. The red line indicates the 5% reduction target.

5%, we consider an experimental design problem where only a subset of houses are treated for long-term vector elimination using locale-specific housing improvements. The algorithm uses a combination of adaptive sampling and Bayesian geostatistical modeling to minimize the number of treated houses while correctly predicting when the reduction target has been met.

**Table 2. Performance of adaptive sampling in the simulation study, compared to random sampling.** Accuracy is the percentage of designs which met the control target, i.e. the out-of-sample infestation rate was below 5% (or 8%). Difference from random is the percentage of the village in the design, minus the corresponding percentage using random sampling, calculated for each initial design.

| Exploration parameter | Covariate set | Accuracy (%) | | Difference (%) median (95% CI) |
|---|---|---|---|---|
| | | 5% target | 8% target | |
| $\alpha = 0$ | Global only | 91.6 | 100 | 4.8 (1.6–11.7) |
| | All | 54.4 | 97.6 | 19.4 (11.2–37.3) |
| $\alpha = 0.15$ | Global only | 97.6 | 100 | 4.8 (2.0–10.6) |
| | All | 52.8 | 97.2 | 18.9 (11.2–36.1) |
| $\alpha = 0.3$ | Global only | 98.0 | 100 | 4.4 (2.0–9.6) |
| | All | 57.2 | 97.2 | 18.4 (11.2–33.7) |
| $\alpha = 0.7$ | Global only | 99.6 | 100 | 4.4 (2.0–7.2) |
| | All | 77.2 | 97.6 | 14.1 (8.4–19.4) |
| $\alpha = 1$ | Global only | 100 | 100 | 4.4 (2.0–7.3) |
| | All | 81.2 | 97.6 | 10.6 (5.6–15.7) |
| $\alpha = 2$ | Global only | 100 | 100 | 2.8 (0–4.1) |
| | All | 94.8 | 100 | 4.4 (1.8–8.4) |
| Random | Global only | 99.2 | 100 | - |
| | All | 98.4 | 100 | - |

The results from our simulation experiment show adaptive sampling strategies are universally more efficient than selecting houses at random, and are able to consistently predict whether the 5% infestation target has been achieved. In the case where only the spatial coordinates of houses are available, even the least exploratory strategy (i.e. $\alpha = 0$) was able to correctly predict the target was met with over 90% accuracy. This is surprising, as selecting houses based on current perceived risk will intuitively lead to biased predictions [32]. One possibility for this is that the spatial information alone leads to minimal shrinkage in the posterior during the initial stages of sampling, which would cause sampling to be nearly random even when $\alpha = 0$ (since unvisited locations will look very similar). A second explanation is that the termination condition we have used counteracts the effects of targeting perceived risk. In particular, sampling in this way can lead to higher variance at unvisited locations since we have neglected to gather relevant information about them, leading to a situation where both the expectation of $\hat{I}_0$ and the probability in (5) are low, and hence avoiding terminating prematurely.

Our second major finding is that including additional socioeconomic covariates can significantly reduce the number of sampled houses, but only for the adaptive sampling strategies. However, this additional information also leads to a greater opportunity for bias in the adaptive strategies with lower exploration parameter, and even larger values of the parameter ($\alpha = 1, 2$) occasionally failed to meet the target, as did random sampling. While this is not ideal, it is reassuring that even with no exploration ($\alpha = 0$), the true infestation rate was below 8% in nearly all of the final designs (Table 2). Since this threshold has been shown to guarantee *T. cruzi* elimination throughout Central America [10], a viable strategy could be to set the target threshold slightly below the level actually desired, thus allowing maximum sampling efficiency while still meeting the actual target.

While the assumption that detailed socioeconomic information is available for the whole community beforehand is not necessarily realistic, the efficiency gained by including these covariates opens up new options for *T. dimidiata* control. For example, this information could be collected beforehand by community members or with a quick survey, thereby reducing the number of homes to search for infestation, which is more time-consuming to collect and requires trained personnel. Moreover, this could allow a more thorough search for Triatomine presence, which is a notoriously noisy measurement [21, 48].

On a practical level, our sampling strategy could compliment the EcoHealth approach in several ways. If local resources for housing improvements are readily available, then adaptive sampling could be used to conserve insecticide and the time of health personnel, while allowing house improvements for all residents who want them. If such resources are scarce, adaptive sampling could further indicate which homes are most in need of immediate improvement. Adaptive sampling could also be used to more quickly develop a localized improvement strategy, by identifying socioeconomic variables most associated with infestation risk from a subsample of relevant houses.

It has been previously suggested that adaptive geostatistical sampling based on prediction variance, followed by targeting areas of higher risk in later stages, could be a way to accurately identify areas for public health interventions while conserving resources [33]. Altogether, our results support this idea, while rigorously comparing different balances of prioritizing variance and risk. Our experiments show that with $\alpha$ sufficiently high, this strategy is robust to different communities and levels of covariate information. The most preferable setting for $\alpha$, however, will depend on the particularities of a given intervention. For example, in the initial stages of an intervention it may be preferable to visit as many communities as possible, with the understanding that the control target may fail to be reached in a few communities, while in later stages a smaller number of remaining areas can be targeted with a higher level of precision.

## 4.1 Full-village analysis

The results from the full-village analysis, where several models were fit to all available data in each of the villages, provide further insight into factors associated with *T. dimidiata* infestation in southeastern Guatemala. First, the superior explanatory power of the spatial models empha-sizes the spatial nature of infestation, something not accounted for in previous studies of infes-tation risk [4, 21]. There are several possible explanations for the spatial autocorrelation present in this data. One possibility is a limited migratory range from existing domestic popu-lations, leading to local clustering following the insect's dispersal season. Such a dynamic is supported by several population genetics studies, which have found insects in nearby houses and adjacent villages are more related [49, 50], as well as insects from highly-infested houses and their neighbors following insecticide application [15]. Another explanation is that spatial patterns are due to unmeasured covariates, which are themselves spatially autocorrelated. While our results show spatial effects remain even after accounting for a number of socioeco-nomic variables, environmental factors such as altitude, temperature, and precipitation have been shown to be mildly correlated with *T. dimidiata* infestation [51, 52].

A second important finding from the full-village analysis is the variation in fixed-effect coefficients among the five villages. While several covariates, such as the condition of bedroom walls, consistently had a meaningful association with infestation, many others varied in their explanatory power, and even whether there was a positive or negative association. Important fixed-effects also varied compared to previous *T. dimidiata* studies in different areas. In partic-ular, the distance from the village perimeter was found to lead to significantly higher infesta-tion rates and vector abundance in Yucatan, Mexico [52]. However, not only was there no correlation ($\rho = 0.02$) between village perimeter and infestation overall in our data, but we found the importance of village perimeter as a covariate was negligible using either covariate set, which shows there is little association between these variables when accounting for spatial and various covariate effects as well. This could be due to the close proximity of several of the villages, or to deforestation leading to a disruption of sylvatic populations in the surrounding environment [53]. Ultimately, these results all add to the existing evidence that risk factors for *T. dimidiata* infestation must be considered in a local context [9, 23].

## 4.2 Limitations and future work

This study has a few limitations. First, our methods focus on bringing the infestation rate below 5% among untreated houses, with the logic that uninfested, unvisited homes already have factors not conducive to *T. dimidiata* infestation, and that the EcoHealth approach empowers communities to maintain these factors throughout the village [22]. However, it is certainly possible for the infestation rate to still rebound. For example, if a previous insecticide application was recent enough, some households may not be uninfested due to favorable con-ditions but rather that the vector population has had insufficient time to fully reestablish. Fur-ther, maintaining certain housing conditions alone may not be enough if changing environmental conditions alter the relationship between housing conditions and infestation risk. Additional research would therefore need to confirm the long-term effects of allowing a small fraction of houses to remain infested after an intervention.

We have also ignored any practical constraints when choosing locations during sampling, such as the travel time to selected points, or the logistics of centralizing incoming data after each iteration of sampling. It therefore may be more realistic to impose a penalty in (4) based on distance from current samples, or to increase the batch size *b* to allow a more natural sam-pling schedule. Finally, our decision to lump all signs of infestation into a single indicator ignores potentially useful information, including their different implications for disease risk. A

recent study in Jutiapa found a large difference between adult and juvenile infestation rates [53], so it may be beneficial to separate these variables.

In the simulation study, several parameters of the adaptive sampling procedure were fixed throughout, such as the batch size $b$ and size of the initial design, to limit computational overhead. While our results demonstrate the settings we chose can be effective across multiple villages, an analysis involving the interaction between these parameters would provide a more thorough summary of when our procedure works best, and its robustness to different parameter settings.

The methods developed in this work can be applied in contexts other than Chagas vector control. In particular, adaptive geostatistical sampling using the utility function (4) and termination condition (5) can be applied nearly directly to other NTD control initiatives seeking to efficiently meet a reduction target. For example, schistosomiasis prevalence in schools has been shown to have spatial autocorrelation, and covariate information for schools can be acquired through teacher-given surveys [54]. Adaptive geostatistical sampling could therefore be applied to efficiently target schools at greater risk.

Our methods could also be applied to other Chagas endemic regions, and extended to account for more complex ecological data, such as jointly modeling several vector species and lifestages, or using spatiotemporal models [55] to improve the efficiency of follow-up surveys and adapt to seasonal effects. It would also be interesting to investigate the use of additional environmental variables as an alternative to the socioeconomic information, which may provide similar gains in sampling efficiency while being easier to collect. More generally, our framework could be used for monitoring multiple, possibly interacting, diseases [56]. This would allow sequential samples to take into account information both between and among pathogens. Finally, additional research should go into the development of tools for adaptive, targeted intervention strategies, and how such software can best be applied for the direct benefit of the community.

In conclusion, we have proposed an adaptive strategy for public health interventions, which transitions from prioritizing areas of greatest uncertainty to those perceived to be most at risk. This would allow control initiatives to use resources more efficiently, by targeting areas of greatest need while still benefiting the entire community. We believe the methods used in this work are well-suited to address the complex ecological, biological, and social factors inherent to disease spread, and hence are applicable to a wide range of epidemiological systems.

## Supporting Information

**S1 Table. Additional information from the 2011 EcoHealth survey.**
(PDF)

**S2 Table. Socioeconomic variables used for model fitting.**
(PDF)

**S1 Appendix. Mathematical definitions.**
(PDF)

**S1 Fig. Effective range from full-village analysis.**
(EPS)

## Acknowledgments

The authors are grateful to the researchers, Ministry of Health personnel, and people of Chiquimula who participated in data collection.

## Author Contributions

**Conceptualization:** B. K. M. Case, Jean-Gabriel Young, Carlota Monroy, Laurent Hébert-Dufresne, Lori Stevens.

**Data curation:** B. K. M. Case, Carlota Monroy.

**Formal analysis:** B. K. M. Case.

**Funding acquisition:** Carlota Monroy, Laurent Hébert-Dufresne, Lori Stevens.

**Investigation:** B. K. M. Case, Carlota Monroy.

**Methodology:** B. K. M. Case, Jean-Gabriel Young, Laurent Hébert-Dufresne.

**Project administration:** B. K. M. Case, Laurent Hébert-Dufresne, Lori Stevens.

**Resources:** Carlota Monroy, Laurent Hébert-Dufresne, Lori Stevens.

**Software:** B. K. M. Case.

**Supervision:** Jean-Gabriel Young, Laurent Hébert-Dufresne, Lori Stevens.

**Validation:** B. K. M. Case.

**Visualization:** B. K. M. Case.

**Writing – original draft:** B. K. M. Case, Jean-Gabriel Young, Laurent Hébert-Dufresne, Lori Stevens.

**Writing – review & editing:** B. K. M. Case, Jean-Gabriel Young, Daniel Penados, Carlota Monroy, Laurent Hébert-Dufresne, Lori Stevens.

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
