## [Decision Letter · Decision Letter 0]

16 Mar 2022

Dear Dr. Hebert-Dufresne,

Thank you very much for submitting your manuscript "Spatial epidemiology and adaptive targeted sampling to manage the Chagas disease vector Triatoma dimidiata" for consideration at PLOS Neglected Tropical Diseases. As with all papers reviewed by the journal, your manuscript was reviewed by members of the editorial board and by several independent reviewers. The reviewers appreciated the attention to an important topic. Based on the reviews, we are likely to accept this manuscript for publication, providing that you modify the manuscript according to the review recommendations. 

Sincerely,

Marilia Sá Carvalho

Associate Editor

Guilherme Werneck

Deputy Editor

Reviewer's Responses to Questions

**Key Review Criteria Required for Acceptance?**

**Methods**

-Are the objectives of the study clearly articulated with a clear testable hypothesis stated?

-Is the study design appropriate to address the stated objectives?

-Is the population clearly described and appropriate for the hypothesis being tested?

-Is the sample size sufficient to ensure adequate power to address the hypothesis being tested?

-Were correct statistical analysis used to support conclusions?

-Are there concerns about ethical or regulatory requirements being met?

Reviewer #1: The presented methods are well described and support the proposed objectives. The methods are well organized and easy to understand. The information was presented in a simple and transparent way.

Yes. The objectives of the study are clearly articulated with the hypothesis.

Yes. The study design is appropriate.

No. The population is not clearly described. More details in the "Summary and General Comments" section.

Yes. The sample size is sufficient to ensure adequate power to address the hypothesis being tested.

Yes. The statistical analysis used to support conclusions was clear.

Reviewer #2: The manuscript is well written, with objectives consistent and consolidated hypothesis.

Reviewer #3: The objectives of the study were clearly articulated with a clear and testable hypothesis. The study design is appropriate to address the exposed goals.

The studied population was sufficiently described in the manuscript and the size in the sample was sufficient for the actual tests.

Good geostatistic solutions were found for the resolution of study questions.

**Results**

-Does the analysis presented match the analysis plan?

-Are the results clearly and completely presented?

-Are the figures (Tables, Images) of sufficient quality for clarity?

Reviewer #1: The results are well presented. The ideas and information presented are sequential and logical. The results are well supported by informative figures and tables. 

Consider better describe the topic "simulation results" and explain the reason why the proposed model "produce designs, which failed to satisfy the design target of reducing the true infestation rate below 5%," and how does it impact a practical application of you idea (*).

The analysis presented matches the analysis plan.

The results are clearly and well presented.

The figures are informative, and in a sufficient quality.

Reviewer #2: The results are clearly and the figures and tables are of sufficient quality for clarity.

Reviewer #3: The results were very well presented and in total sync with the objectives of the study.

**Conclusions**

-Are the conclusions supported by the data presented?

-Are the limitations of analysis clearly described?

-Do the authors discuss how these data can be helpful to advance our understanding of the topic under study?

-Is public health relevance addressed?

Reviewer #1: The article suggests new options for implementing long-term T. dimidiata vector control. Describe an adaptive strategy for public health interventions, which transitions from prioritizing areas of greatest uncertainty to those perceived to be most at risk, using data from five villages in southeastern Guatemala.

This is an interesting approach bringing relevant strategies aiming to interrupt disease transfer, targeting only a subset of house community. Also, demonstrate how sampling strategies could compliment the EcoHealth initiative to conserve insecticide, the time of health personnel and other important approaches.

The conclusions are supported by the data presented.

The limitations of analysis are clearly described.

The authors discuss well how these data can be helpful.

Reviewer #2: The manuscript "Spatial epidemiology and adaptive targeted sampling to manage the Chagas disease vector Triatoma dimidiata" brings some important challenges for entomological surveillance, as the operational capacity of the service, to reduce the proportion of infected households with Triatoma dimidiata in short time.

Reviewer #3: The conclusions of the study were correctly placed. Do not extrapolate the capacity of the study. Leaving the goal of the study to be another useful tool in combating Chagas in small communities.

The authors were also quite clear in the study limitations and the need for new developments and approaches.

**Editorial and Data Presentation Modifications?**

Reviewer #1: Specific Comments

Study population and sample collection

The manuscript need to better describe the study population and the sampling methods; 

The manuscript should inform the proportion of "missing data" removed from the study, for better understanding of readers.(*)

Taking into consideration the analysis of this proportion by each location. 

Line 8 – Suggestion of change: Triatomine levels – triatomine infestation levels. 

Line 11 – Better develop the idea of seroprevalence in this sentence. Is this the seroprevalence of domestic animals, of humans? Please, be more informative.

Line 16 – Suggestion of change: … is endemic to the continent and infests peridomestic and sylvatic environments … – … is endemic to the continent, living at sylvatic and infesting peridomestics (domestic?) environments…

Line 114 – The manuscript does not explain numbers of “missing information”  and how much it represents the total and by each location;

Line 118 – Do the team member follow some specific protocol to search the houses and find the T. cruzi vectors?

Reviewer #2: (No Response)

Reviewer #3: First I would like to congratulate the work presented. Demonstrated methodological clarity and alignment with the very significant goals. It was also written very clearly.

That said I would like to put some suggestions that I believe can enrich work or future unfolding.

The authors selected socioeconomic variables all linked to the domicile and perehyde. Surely the use of indicators expressing the ecological condition of the surroundings of households and the village will erich the work.

The vectors despite being well adapted to the human domicile can have in the external areas of the domicile and the ecological niches that operate from reservoirs for future infestations. In addition to the ecological indicators I believe that the description and the preparation of cultural indicators of the locations and residents of households would be much relevant. They are certainly difficult to operational.

In both cases if it is not possible to controll the future study of these types of ecological and cultural indicators that would approach the study of the theoretical reference presented in it. That at least make a description of the ecological framework and culture from locations such as percentage of vegetated area, rainfall etc.

Finally I suggest that they approach the environmental problems intriscated the use of insetiside to the environment as well as the selection of resistant individuals in the veonstial population.

**Summary and General Comments**

Reviewer #1: The main objective of this research was duly achieved, the methodologies and results are duly described and discussed. Congratulations. Despite that, as a neglected population disease, it is important to bring relevant techniques, such as those presented by the authors in this publication, to governments and vulnerable communities accessibly. As a suggestion, evaluate the possibility of including in the discussion of this manuscript the necessity of developing a tool (system or application), widely accessible, to help regional leaders and government to implement spatial geoestatitical models, helping mitigate the reality of these communities, in a more accessible way. Alternatively, suggesting the validation of the proposed methods in other countries will help achieve breadth and certification of the methods widely.

Specific Comments

Study population and sample collection

The manuscript need to better describe the study population and the sampling methods; 

The manuscript should inform the proportion of "missing data" removed of the study, for better understanding of readers.(*)

Taking into consideration the analysis of this proportion by each location. 

Line 8 – Suggestion of change: Triatomine levels – triatomine infestation levels. 

Line 11 – Better develop the idea of seroprevalence in this sentence. Is this the seroprevalence of domestic animals, of humans? Please, be more informative.

Line 16 – Suggestion of change: … is endemic to the continent and infests peridomestic and sylvatic environments … – … is endemic to the continent, living at sylvatic and infesting peridomestics (domestic?) environments…

Line 114 – The manuscript does not explain numbers of “missing information”  and how much it represents the total and by each location;

Line 118 – Do the team member follow some specific protocol to search the houses and find the T. cruzi vectors?

Reviewer #2: (No Response)

Reviewer #3: (No Response)

PLOS authors have the option to publish the peer review history of their article (what does this mean?). If published, this will include your full peer review and any attached files.

Reviewer #1: No

Reviewer #2: No

Reviewer #3: No

Figure Files:

Data Requirements:

Reproducibility:

References

---

## [Editor Report · Decision Letter 1]

20 Apr 2022

Dear Dr. Hebert-Dufresne,

We are pleased to inform you that your manuscript 'Spatial epidemiology and adaptive targeted sampling to manage the Chagas disease vector Triatoma dimidiata' has been provisionally accepted for publication in PLOS Neglected Tropical Diseases.

Best regards,

Marilia Sá Carvalho

Associate Editor

Guilherme Werneck

Deputy Editor

---

## [Editor Report · Acceptance letter]

16 May 2022

Dear Mx. Case,

We are delighted to inform you that your manuscript, "Spatial epidemiology and adaptive targeted sampling to manage the Chagas disease vector Triatoma dimidiata," has been formally accepted for publication in PLOS Neglected Tropical Diseases.

Best regards,

Shaden Kamhawi

co-Editor-in-Chief

Paul Brindley

co-Editor-in-Chief
